# Neural Information Squeezer for Causal Emergence

**DOI:** 10.3390/e25010026

**Published:** 2022-12-23

**Authors:** Jiang Zhang, Kaiwei Liu

**Affiliations:** 1School of Systems Sciences, Beijing Normal University, Beijing 100875, China; kevinliunxt@163.com; 2Swarma Research, Beijing 100085, China

**Keywords:** causal emergence, coarse-graining, invertible neural network

## Abstract

Conventional studies of causal emergence have revealed that stronger causality can be obtained on the macro-level than the micro-level of the same Markovian dynamical systems if an appropriate coarse-graining strategy has been conducted on the micro-states. However, identifying this emergent causality from data is still a difficult problem that has not been solved because the appropriate coarse-graining strategy can not be found easily. This paper proposes a general machine learning framework called Neural Information Squeezer to automatically extract the effective coarse-graining strategy and the macro-level dynamics, as well as identify causal emergence directly from time series data. By using invertible neural network, we can decompose any coarse-graining strategy into two separate procedures: information conversion and information discarding. In this way, we can not only exactly control the width of the information channel, but also can derive some important properties analytically. We also show how our framework can extract the coarse-graining functions and the dynamics on different levels, as well as identify causal emergence from the data on several exampled systems.

## 1. Introduction

Emergence, as one of the most important concepts in complex systems, describes the phenomenon that some overall properties of a system cannot be reduced to the parts [1,2]. Causality, as another significant concept, characterises the connection between cause and effect events through time [3,4] for a dynamical system. As pointed out by Hoel et al. [5,6], causality could be emergent, which means that the events of a system on the macro level may have stronger causal connections than the micro level, where the strength of causality could be measured by effective information (EI) [5,7]. This theoretical framework of causal emergence provides us a new way to understand emergence and other important conceptions in a quantitative way [8,9,10,11]. Previous works have shown that causal emergence can be applied in wide areas including studies on ant colony [12], protein interactomes [13], brain [14], and biological networks [15].

Although many concrete examples of causal emergence across different temporal and spatial scales have been shown in [5], a method to identify causal emergence merely from data is needed. This problem is hard because a method to search all possible coarse-graining strategies (functions, mappings) in a systematic and automatic way, in which the causal emergence can be shown [10], is needed.

The problem is difficult because the search space is all possible mapping functions between micro and macro, which is huge. To solve this problem, Klein et al. focuses on the complex systems with network structures [15,16], and converted the problem of coarse-graining into node clustering. That is, to find a way to group nodes into clusters such that the connections on the cluster level has larger EI than the original network. This method has been widely applied in various areas [12,13,14], nevertheless, it assumes that the underlying node dynamic is diffusion (random walks). Meanwhile, real complex systems have much richer node dynamics. For a general dynamic, even if the node grouping is given, the coarse-grained strategy still needs to consider how to map the micro-states of all nodes in a cluster to the macro-state of the cluster [5]. The tedious searching on a huge functional space of coarse-graining strategies is also needed.

When we consider all possible mappings, another difficulty is trivial coarse-graining strategies avoiding. An exampled trivial method is to map all the micro-states into an identical value as the macro-state. In this way, the macroscopic dynamics is only an identical mapping that will have large effective information (EI) measure. However, this can not be called causal emergence because all the information is eliminated by the coarse-graining method itself. Thus, we must find a way to exclude such trivial strategies.

An alternative way to identify causal emergence from data is based on partial information decomposition given by [10,17]. Although the methods based on information decomposition can avoid the discussion on coarse-graining strategies, a time consuming search on subsets of the system state space is also needed if we want to obtain the exact value. In addition, the reported numeric approximate method can only provide sufficient condition. Further, the method can not give the explicit coarse-graining strategy and the corresponding macro-dynamics which are useful in practice. Another common shortage shared by the two mentioned methods is that an explicit Markov transition matrix for both macro- and micro-dynamics are needed, and the transitional probabilities should be estimated from data. As a result, large bias on rare events can hardly be avoided, particularly for continuous data.

On the other hand, machine learning methods empowered by neural networks have been developed in recent years, and many cross-disciplinary applications have been made [18,19,20,21]. Equipped with this method, automated discovery of causal relationships and even dynamics of complex systems in a data driven way becomes possible [22,23,24,25,26,27,28,29]. Machine learning and neural networks can also help us to find good coarse-graining strategies [30,31,32,33,34]. If we treat a coarse-graining mapping as a function from micro-states to macro-states, then we can certainly approximate this function by a parameterized neural network. For example, Refs. [31,33] used normalized flow model equipped with invertible neural network to learn how to renormalize a multi-dimensional field (quantum field, images or joint probability distributions), and how to generate the field from Gaussian noise. Therefore, both the coarse-graining strategy and the generative model can be learned from data automatically.

These techniques can also help us to reveal causality on macro-level from data. Causal representation learning aims to use unsupervised representation learning to extract causal latent variables behind the observational data [35,36]. The encoding process from the original data to the latent causal variables can be understood as a kind of coarse-graining. This shows the similarity between causal emergence identification and causal representation learning; however, their basic objectives are different. Causal representation learning aims to extract the causality hidden in data, whereas causal emergence identification aims to find a good strategy of coarse-graining to reduce the given micro-level dynamics. Furthermore, introducing multi-scale modeling and coarse-graining operations into causal models brings some new theoretical problems [37,38,39]. For example, Refs. [38,39] discuss the basic requirements of the model abstraction (coarse-graining). However, these studies only care about static random variables and structural causal models but not Markovian dynamics.

In this paper, we formulate the problem of causal emergence identification as a maximization problem of the effective information (EI) for the macro-dynamics under the constraint of precise prediction of micro-dynamics. We then propose a general machine learning framework called Neural Information Squeezer (NIS) to solve the problem. By using invertible neural network to model the coarse-graining strategy, we can decompose any mapping from 
$R^{p}$
 to 
$R^{q}$
 (
q≤p
) into a series of information conversions invertible processes and information discarding processes. In this way, the framework can not only allow us to control information conversion and discarding in a precise way but also enable us to mathematically analyze the whole framework in theory. We prove a series of mathematical theorems to reveal the properties of NIS. At last, we show how NIS can learn effective coarse-graining strategies and macro-state dynamics numerically on a set of examples.

## 2. Basic Notions and Problems Formulation

First, we will formulate our problems under a general setting, and layout our framework to solve the problems.

### 2.1. Background

Suppose the dynamics of the complex system that we consider can be described by a set of differential equations.

(1)
$$\frac{dx}{dt}=g\left(x(t),\xi \right),$$

where 
$x\left(t\right)\in R^{p}$
 is the state of the system and 
$p\in Z^{+}$
 is a positive integer, 
ξ
 is a gaussian random noise. Normally, micro-dynamic *g* is always Markovian which means it could be also modeled as a conditional probability 
Pr(x(t+dt)|x(t))
 equivalently.

However, we cannot directly obtain the evolution of the system but the discrete samples of the states, and we define these states as micro-states.

**Definition** **1.***(Micro-states): Each sample of the state of the dynamical system (Equation* (Equation 1)*) 
$x_{t}$
 is called a micro-state at time step t. In addition, the multi-variate time series 
$x_{1},x_{2},\cdot \cdot \cdot ,x_{T}$
 which are sampled with equal intervals and a finite time step T, forms a micro-state time series.*

We always want to reconstruct *g* according to the observable micro-states. However, an informative dynamical mechanism *g* with strong causal connections is always hard to be reconstructed from the micro-states when noise 
ξ
 is strong. Meanwhile, we can ignore some information in the micro-state data and convert it into macro-state time series. In this way, we may reconstruct a macro-dynamic with stronger causality to describe the evolution of the system. This is the basic idea behind causal emergence [5,6]. We formalize the information ignoring process as a coarse-graining strategy (or mapping, method).

**Definition** **2.**
*(q dimensional coarse-graining strategy): Suppose the dimension of the macro-states is 
$0<q<p\in Z^{+}$
, a q dimensional coarse-graining strategy is a continuous and differential function to map the micro-state 
$x_{t}\in R^{p}$
 to a macro-state 
$y_{t}\in R^{q}$
. The coarse-graining is denoted as 
$\phi_{q}$
.*


After coarse-graining, we obtain a new time series data of macro-states denoted by 
$y_{1}=\phi_{q}\left(x_{1}\right),y_{2}=\phi_{q}\left(x_{2}\right),\cdot \cdot \cdot ,y_{T}=\phi_{q}\left(x_{T}\right)$
. We then try to find another dynamical model (or a Markov chain) 
${\hat{f}}_{\phi_{q}}$
 to describe the evolution of 
$y_{t}$
:

**Definition** **3.***(macro-state dynamics): For the given time series of macro-states 
$y_{1},y_{2},\cdot \cdot \cdot ,y_{T}$
, the macro-state dynamics are a set of differential equations*

(2)
$$\frac{dy}{dt}={\hat{f}}_{\phi_{q}}\left(y,\xi^{\prime }\right),$$

*where 
$y\in R^{q}$
, 
$\xi^{\prime }\in R^{q}$
 is the gaussian noise in the macro-state dynamics, and 
${\hat{f}}_{\phi_{q}}$
 is a continuous and differential function such that the solution of Equation* (Equation 2)*, 
y(t)
, can minimize:*

(3)
$$\langle ||y_{t}-{y\left(t\right)||\rangle }_{\xi^{\prime }}$$

*for any given time step 
t∈[1,T]
 and given vector form 
||·||
.*

However, this formulation cannot reject some trivial strategies. For example, suppose a 
q=1
 dimensional 
$\phi_{q}$
 is defined as 
$\phi_{q}\left(x_{t}\right)=1$
 for 
$\forall y_{t}\in R^{p}$
. Thus, the corresponding macro-dynamic is simply 
dy/dt=0
 and 
y(0)=1
. However, this is meaningless because the macro-state dynamic is trivial and coarse-graining mapping is too arbitrary.

Therefore, we must set limitations on coarse-graining strategies and macro-dynamics so that such trivial strategies and dynamics could be avoided.

### 2.2. Effective Coarse-Graining Strategy and Macro-Dynamics

We define an effective coarse-graining strategy to be a compressed map such that the macro-states may preserve the information of micro-states as much as it can. Formally,

**Definition** **4.***(ϵ-effective q coarse-graining strategy and macro-dynamcis): A q coarse-graining strategy 
$\phi_{q}:R^{p}\to R^{q}$
 is ϵ-effective (or abbreviate as effective) if there exists a function 
$\phi_{q}^{†}:R^{q}\to R^{p}$
, such that the following inequality holds for a given small real number ϵ and given vector norm 
||·||
:*

(4)
$$\left|\right|\phi_{q}^{†}\left(y\left(t\right)\right)-x_{t}\left|\right|<\epsilon ,$$

*and the derived macro-dynamic 
${\hat{f}}_{\phi_{q}}$
 is also ϵ-effective, where 
y(t)
 is the solution of Equation* (Equation 2)*, that is:*

(5)
$$y\left(t\right)=\phi_{q}\left(x_{t-1}\right)+\int_{t-1}^{t}{\hat{f}}_{\phi_{q}}\left(y\left(\tau \right),\xi^{\prime }\right)d\tau$$

*for all 
t=1,2,...,T
. Thus, we can reconstruct the micro-state time series by 
$\phi_{q}^{†}$
 such that the macro-state variables contain the information of micro-states as much as they can.*

Notice that this definition is in accordance with the approximate causal model abstraction [40].

### 2.3. Problem Formulation

Our final objective is to find a most informative macro-dynamic. Therefore, we need to optimize the coarse-graining strategy and the macro-dynamic among all possible effective strategies and dynamics. Therefore, our problem can be formulated as:
(6)
$$\max_{\phi_{q},{\hat{f}}_{\phi_{q}},\phi_{q}^{†},q}I\left({\hat{f}}_{\phi_{q}}\right),$$

under the constraint Equations (Equation 4) and (Equation 5), where 
I
 is a measure of effective information, it could be 
EI
, 
Eff
, or dimension averaged EI which is mainly used in this paper and is denoted as 
dEI
 (will mention in Section 3.3.3). 
$\phi_{q}$
 is an effective coarse-graining strategy, and 
${\hat{f}}_{\phi_{q}}$
 is an effective macro-dynamic.

## 3. Methods

The problem (Equations (Equation 6) and (Equation 4)) is hard to solve because the objects that we will optimize are functions: 
$\phi_{q},{\hat{f}}_{\phi_{q}},\phi_{q}^{†}$
 but not numbers. Thus, we use neural networks to parameterize the functions and convert the function optimization problem into a parameter optimization problem.

### 3.1. Neural Information Squeezer Model

We propose a new machine learning framework called neural information squeezer (NIS) which is based on invertible neural network to solve the problem (Equation (Equation 6)). NIS is composed of three components: encoder, dynamics learner, and decoder. They are represented by neural networks 
$\psi_{\alpha }$
, 
$f_{\beta }$
, and 
$\psi_{\alpha }^{-1}$
 with the parameters 
α,β
, and 
α
 respectively. The entire framework is shown in Figure 1. Next, we will describe each module separately.

#### Encoder

To be noticed, 
$\psi_{\alpha }$
 is an invertible neural network (INN), therefore 
ψ
 and 
$\psi^{-1}$
 share the parameters 
α
. However, invertible function has no information loss, we must introduce a new operator, projection.

**Definition** **5.***(Projection operator): A projection operator 
$\chi_{p,q}$
 is a function from 
$R^{p}$
 to 
$R^{q}$
, such that:*

(7)
$$\chi_{p,q}\left(x_{q}⨁x_{p-q}\right)=x_{q},$$

*where ⨁ is the operation of vector concatenation, and 
$x_{q}\in R^{q},x_{p-q}\in R^{p-q}$
. Sometimes, we abbreviate 
$\chi_{p,q}$
 as 
$\chi_{q}$
 if there is no ambiguity.*

Thus, the encoder (
ϕ
) maps the micro-state 
$x_{t}$
 to the macro-state 
$y_{t}$
, and this mapping can be separated into two steps. That is,

(8)
$$\phi_{q}=\chi_{q}\circ \psi_{\alpha },$$

where ∘ represents the operation of function composition.

The first step is a bijective (invertible) mapping 
$\psi_{\alpha }:R^{p}\to R^{p}$
 from 
$x_{t}\in R^{p}$
 to 
$x_{t}^{\prime }\in R^{p}$
 without information lose and is realized by an invertible neural network, the second step is to project the resulting vector to *q* dimension by mapping 
$x_{t}^{\prime }\in R^{p}$
 into 
$y_{t}\in R^{q}$
 by discarding the information on 
p−q
 dimension.

There are several ways to realize an invertible neural network [41,42]. Meanwhile, we select RealNVP module [43] as shown in Figure 2 to concretely implement the invertible computation.

In the module, the input vector 
x
 can be separated into two parts, both vectors will be scaled, translated and merged again. The magnitude of the scaling and translation operations will be adjusted by the corresponding feed-forward neural networks. 
$s_{1},s_{2}$
 are the same neural networks shared parameters for scaling, ⨂ represents element-wised product. In addition, 
$t_{1},t_{2}$
 are the neural networks shared parameters for translation. In this way, an invertible computation from 
x
 to 
y
 can be realized. The same module can be repeated for multiple times (three times in this paper) to realize complex invertible computation, the details can be referred to Appendix A.

The reasons why we use invertible neural network are: (1) INN can reduce the complexity of the model by multiplexing the structure and the parameters in the encoder to the decoder because we can simply reverse the running direction of the encoder to implement decoding; (2) the encoder equipped with INN can separate out the information conversion process and information discarding process; (3) this enables us to do mathematical analysis on the whole framework, and several theorems reflecting the basic properties can be proved.

### 3.2. Decoder

The decoder converts the predicted macro-state of the next time step 
y(t+1)
 into the prediction of the micro-state at the next time step 
${\hat{x}}_{t+1}$
. In our framework, because the coarse-graining strategy 
$\phi_{q}$
 can be decomposed as a bijector 
$\psi_{\alpha }$
 and a projector 
$\chi_{q}$
, we can simply reverse 
$\psi_{\alpha }$
 to become 
$\psi_{\alpha }^{-1}$
 as the decoder. However, because the dimension of the macro-state is *q* and the input dimension of 
$\psi_{\alpha }$
 is 
p>q
, we need to fill the remaining 
p−q
 dimensions by a 
p−q
 dimensional Gaussian random vector. That is, for any 
$\phi_{q}$
, the decoding mapping can be defined as:
(9)
$$\phi_{q}^{†}=\psi_{\alpha }^{-1}\circ \chi_{q}^{†},$$

where 
$\psi_{\alpha }^{-1}$
 is the inverse function of 
$\psi_{\alpha }$
, and 
$\chi_{q}^{†}:R^{q}\to R^{p}$
 is a function defined as follow: for any 
$x_{q}\in R^{p}$


(10)
$$\chi_{q}^{†}\left(x_{q}\right)=x_{q}⨁z_{p-q},$$

where 
$z_{p-q}∼N\left(0,I_{p-q}\right)$
 is a random Gaussian noise with 
p−q
 dimension, and 
$I_{p-q}$
 is an identity matrix with the same dimension. That is, we can generate a micro-state by composing 
$x_{q}$
 and a random sample 
$z_{p-q}$
 from a 
p−q
 dimensional standard normal distribution.

According to the point view of [31,33], the decoder can be regarded as a generative model of the conditional probability 
$Pr({\hat{x}}_{t+1}|y\left(t+1\right))$
, and the encoder just performs a renormalization process.

#### Dynamics Learner

The dynamics learner 
$f_{\beta }$
 is a common feed-forward neural network with parameters 
β
, it will learn the effective Markov dynamic on the macro-level. Concretely, we at first use 
$f_{\beta }$
 to replace 
${\hat{f}}_{\phi_{q}}$
 in Equation (Equation 2), and second we use Euler method with 
dt=1
 to solve the Equation (Equation 2), and suppose the noise is a additive Gaussian (or Laplacian) [44], therefore we can reduce Equation (Equation 5) as:
(11)
$$y\left(t+1\right)=y_{t}+\int_{t}^{t+1}f_{\beta }\left(y\left(\tau \right),\xi^{\prime }\right)d\tau \approx y_{t}+f_{\beta }\left(y_{t}\right)+\xi^{\prime }$$

where 
$\xi^{\prime }∼N\left(0,\Sigma \right)$
 or 
Laplacian(0,Σ)
, 
$\Sigma =diag(\sigma_{1}^{2},\sigma_{2}^{2},\cdot \cdot \cdot ,\sigma_{q}^{2})$
 is the covariance matrix, and 
$\sigma_{i}$
 is the standard deviation in the *i*th dimension which could be learned or fixed. Thus, the transitional probability of this dynamics can be written as

(12)
$$P\left(y\left(t+1\right)|y_{t}\right)=D\left(\mu \left(y_{t}\right),\Sigma \right),$$

where 
D
 represents the PDF of Gaussian distribution or Laplace distribution, 
$\mu \left(y_{t}\right)\equiv y_{t}+f_{\beta }\left(y_{t}\right)$
 is the mean vector of the distribution.

By training the dynamics learner in an end-to-end manner, we can avoid estimating the Markov transitional probabilities from the data to reduce biases because neural networks always have much better ability to fit the data and generalize to unseen cases.

### 3.3. Two Stage Optimization

Although the functions that will be optimized have been parameterized by neural networks, Equation (Equation 6) is still hard to be optimized directly because the objective function and the constraint condition must be combined together to be considered and *q* as a hyper-parameter can affect the structure of neural networks. Thus, in this paper, we propose a two-stage optimization method. In the first stage, we fix the hyper-parameter *q* and optimize the difference between the predicted micro-state and the observed data 
$|\phi_{q}^{†}\left(y\left(t\right)\right)-x_{t}|$
, that is Equation (Equation 4), to let the coarse-graining strategy 
$\phi_{q}$
 and macro-dynamics 
${\hat{f}}_{q}$
 to be effective. Then, we search for all possible *q* values to find the optimal one such that 
I
 can be maximized.

#### 3.3.1. Stage 1: Training a Predictor

In the first stage, we can use likelihood maximization and stochastic gradient descend techniques to obtain the effective *q* coarse-graining strategy and the effective predictor of the macro-state dynamics. The objective function is defined on the likelihood of micro-state prediction.

We can understand a feed-forward neural network as a machine to model a conditional probability with Gaussian or Laplacian distribution [44]. Thus, the entire NIS framework can be understood as a model of 
$P({\hat{x}}_{t+dt}|x_{t})$
 with the output 
${\hat{x}}_{t+1}$
 is just the mean value. In addition, the objective function Equation (Equation 14) is just the log-likelihood or cross-entropy of the observed data under the given form of the distribution.

(13)
$$L=\sum_{t}\ln P\left({\hat{x}}_{t+1}=x_{t+1}|x_{t}\right),$$

where 
$P\left({\hat{x}}_{t+1}=x_{t+1}|x_{t}\right)\equiv N\left({\hat{x}}_{t+1},\Sigma \right)$
 when 
l=2
 or 
$Laplace({\hat{x}}_{t+1},\Sigma )$
 when 
l=1
, where 
Σ
 is the covariance matrix which is always be a diagonal matrix and the magnitude can be calculated as the mean square error for 
l=2
 or mean absolute value for 
l=1
.

If we take the concrete form of Gaussian or Laplacian distribution into the conditional probability, we will see to maximize the log-likelihood is equivalent to minimize the *l*-norm objective function:
(14)
$$L=\sum_{t}\left|\right|{\hat{x}}_{t+1}-x_{t+1}{\left|\right|}_{l}$$

where 
l=1
 or 2.

Then, we can use stochastic gradient descend technique to optimize Equation (Equation 14).

#### 3.3.2. Stage 2: Search for the Optimal Scale

In the previous step, we can obtain the effective *q* coarse graining strategy and the macro-state dynamics after a large number of training epochs, but the results are dependent on *q*.

To select the optimized *q*, we can compare the measure of effective information 
I
 for different *q* coarse-graining macro-dynamics. Because the parameter *q* only has one dimension, and its value range is also limited (
0<q<p
), we can simply iterate all *q* to find out the optimal 
$q^{*}$
 and the optimal effective strategy.

#### 3.3.3. About Effective Information

In the second stage, to compare coarse-graining strategies and macro-dynamics, we need to compute the important indicator: effective information (EI); however, the conventional computations of EIs are all for discrete Markov dynamics in most of previous works [5,6], and we may confront difficulties when we apply EI on continuous dynamics [9].

First, the conventional methods on mutual information computation for discrete variables cannot be used here, new methods for continuous variables and mappings especially for high dimensional space must be invented. To solve the problem, we treat the mapping of the dynamics learner neural network as an conditional Gaussian distribution, thereafter, we can calculate EI for this Gaussian distribution. Concretely, we have the following theorem:

**Theorem** **1.***(EI for feed-forward neural networks) In general, if the input of a neural network is 
$X=\left(x_{1},x_{2},\cdot \cdot \cdot ,x_{n}\right)\in {\left[-L,L\right]}^{n}$
, which means X is defined on a hyper-cube with size L, where L is a very large integer. The output is 
$Y=(y_{1},y_{2},\cdot \cdot \cdot ,y_{m})$
, and 
Y=μ(X)
. Here μ is the deterministic mapping implemented by the neural network: 
$\mu :R^{n}\to R^{m}$
, and its Jacobian matrix at X is 
$\partial_{X^{\prime }}\mu \left(X\right)\equiv {\left\{\frac{\partial \mu_{i}\left(X^{\prime }\right)}{\partial X_{j}^{\prime }}{|}_{X^{\prime }=X}\right\}}_{nm}$
. If the neural network can be regarded as a Gaussian distribution conditional on given X:*

(15)
$$p\left(Y|X\right)=\frac{1}{\sqrt{{\left(2\pi \right)}^{m}\left|\Sigma \right|}}\exp\left(-\frac{1}{2}{\left(Y-\mu \left(X\right)\right)}^{T}\Sigma^{-1}\left(Y-\mu \left(X\right)\right)\right)$$

*where 
$\Sigma =diag(\sigma_{1}^{2},\sigma_{2}^{2},\cdot \cdot \cdot ,\sigma_{m}^{2})$
 is the co-variance matrix, and 
$\sigma_{i}$
 is the standard deviation of the output 
$y_{i}$
 which can be estimated by the mean square error of 
$y_{i}$
, then the effective information (EI) of the neural network can be calculated in the following way:**(i) If there exists X such that 
$\det(\partial_{X^{\prime }}\mu \left(X\right))\neq 0$
, then the effective information (EI) can be calculated as:*

(16)
$$\begin{array}{cc} EI_{L}\left(\mu \right)=I(do(X∼U\left({\left[-L,L\right]}^{n};Y\right)\approx & -\frac{m+m\ln\left(2\pi \right)+\sum_{i=1}^{m}\sigma_{i}^{2}}{2}\\ \; & +n\ln\left(2L\right)+E_{X∼U({\left[-L,L\right]}^{n}}\left(\ln|\det(\partial_{X^{\prime }}\mu \left(X\right))|\right). \end{array}$$

*where 
$U({\left[-L,L\right]}^{n})$
 is the uniform distribution on 
${\left[-L,L\right]}^{n}$
, and 
|·|
 is absolute value, and det is determinant.*
*(ii) If 
$\det(\partial_{X^{\prime }}\mu \left(X\right))\equiv 0$
 for all X, then 
EI≈0
.*


Although Theorem 1 can solve the problem of EI computation for continuous variables and functions, new problems must be confronted which are: (1) EI will be affected by the output dimension *m* easily, this may trouble the comparison of EI for different dimensional dynamics, and (2) EI is dependent on *L*, and will be divergent when *L* is very large.

To solve the first problem, we define a new indicator which is called dimension averaged effective information or effective information per dimension. Formally,

**Definition** **6.***(Dimension Averaged Effective Information (dEI)): For a dynamic f with n dimensional state space, then the dimension averaged effective information is defined as:*

(17)
$$dEI\left(f\right)=\frac{EI(f)}{n}$$


Therefore, if the dynamic *f* is continuous and can be regarded as a conditional Gaussian distribution, then according to Theorem 1, the dimension averaged EI can be calculated as (
m=n
):
(18)
$$dEI_{L}\left(f\right)=-\frac{1+\ln\left(2\pi \right)+\sum_{i=1}^{n}\sigma_{i}^{2}/n}{2}+\ln\left(2L\right)+\frac{1}{n}E_{X∼U({\left[-L,L\right]}^{n}}\left(\ln|\det(\partial_{X^{\prime }}f\left(X\right))|\right).$$


It is easy to see that all the terms related with dimension *n* in Equation (Equation 18) is eliminated. However, there is still *L* in the equation which may cause divergent when *L* is very large.

Therefore, to solve this problem, we can calculate the dimension averaged causal emergence (dCE) to eliminate the influence of *L*.

**Definition** **7.***(Dimension averaged causal emergence (dCE)): for macro-dynamics 
$f_{M}$
 with dimension 
$n_{M}$
 and micro-dynamics 
$f_{m}$
 with dimension 
$n_{m}$
, we define dimension averaged causal emergence as:*

(19)
$$dCE\left(f_{M},f_{m}\right)=dEI\left(f_{M}\right)-dEI\left(f_{m}\right)=\frac{EI(f_{M})}{n_{M}}-\frac{EI(f_{m})}{n_{m}}.$$


Thus, if the dynamics 
$f_{M}$
 and 
$f_{m}$
 are continuous and can be regarded as conditional Gaussian distributions, then according to Definition 7 and Equation (Equation 18), the dimension averaged causal emergence can be calculated as:
(20)
$$\begin{array}{cc} dCE(f_{M},f_{m})= & \left(\frac{1}{n_{M}}E_{X_{M}}\ln|\det\partial_{X_{M}}f_{M}|-\frac{1}{n_{m}}E_{X_{m}}\ln\left|\det\partial_{X_{m}}f_{m}\right|\right)\\ \; & -\left(\frac{1}{n_{M}}\sum_{i=1}^{n_{M}}\ln\sigma_{i,M}^{2}-\frac{1}{n_{m}}\sum_{i=1}^{n_{m}}\ln\sigma_{i,m}^{2}\right) \end{array}$$


Therefore, all the effects of dimension *n* and *L* have been eliminated in Equation (Equation 20), and the result is only influenced by the relative values of the variances and the logarithmic values of the determinant of the jacobian matrices. In the following numeric computations, we will mainly use Equation (Equation 20). The reason why we not use Eff is also because it contains *L*.

## 4. Results

In this section, we will layout several theoretic properties of NIS at first; then, we will apply it on some numeric examples.

### 4.1. Theoretical Analysis

To understand why the neural information squeezer framework can find out the most informative macro-dynamics and how the effective strategy and dynamics change with *q*, we at first layout some major theoretical results through mathematical analysis. Notice that although all of the theorems are about mutual information, these conclusions are also suitable for effective information because all the theoretical results are irrelevant to the distribution of input data.

#### 4.1.1. Squeezed Information Channel

First, we notice that the framework (Figure 1) can be regarded as an information channel as shown in Figure 3, and due to the existence of the projection operation, the channel is squeezed in the middle. Therefore, we call that a squeezed information channel (see also Appendix B for formal definition for the sqeezed information channel).

As proved in Appendix B, we have a theorem for the squeezed information channel:

**Theorem** **2.***(Information bottleneck of Information Squeezer): For the squeezed information channel as shown in Figure 3 and for any bijector ψ, projector 
$\chi_{q}$
, macro-dynamics f, and the random noise 
$z_{p-q}∼N\left(0,I_{p-q}\right)$
, we have:*

(21)
$$I\left(y_{t};y\left(t+1\right)\right)=I\left(x_{t};{\hat{x}}_{t+1}\right),$$

*where 
${\hat{x}}_{t+1}$
 is the prediction of NIS, and 
y(t+1)
 follows Equation* (Equation 2)*.*

That is, for any neural network that implements the general framework as shown in Figure 3, the mutual information of macro-dynamic 
$f_{\phi_{q}}$
 is identical to the entire dynamical model, i.e., the mapping from 
$x_{t}$
 to 
${\hat{x}}_{t+1}$
 for any time. Theorem 2 is fundamental for NIS. Actually, the macro-dynamics *f* is the information bottleneck of the entire channel [45].

#### 4.1.2. What Happens during Training

With Theorem 2, we can understand what happens when the neural squeezer framework is trained by data in an intuitive way.

First, we know as the neural networks are trained, the output of the entire framework 
${\hat{x}}_{t+1}$
 is closed to the real data 
$x_{t+1}$
 under any given 
$x_{t}$
, so do the mutual information, that is the following theorem:

**Theorem** **3.***(Mutual information of the model will be closed to the data for a well trained framework): If the neural networks in NIS framework are well-trained (which means for any 
t∈[1,T]
, the Kullback-Leibler divergence between 
$Pr_{\tau }\left({\hat{x}}_{t+1}|x_{t}\right)$
 and 
$Pr(x_{t+1}|x_{t})$
 approaches 0 when the training epoch 
τ→∞
), then:*

(22)
$$I\left({\hat{x}}_{t+1};x_{t}\right)≃I\left(x_{t+1};x_{t}\right)$$

*for any 
t∈[1,T]
, where ≃ means asymptotic equivalence when 
τ→∞
.*

The proof is in the Appendix C.

Second, we suppose that the mutual information 
$I(x_{t},x_{t+1})$
 is always large because the time series of micro-states 
$x_{t}$
 contains information. Otherwise, we may not be interested in 
$\left\{x_{t}\right\}$
. Therefore, as the neural network is trained, 
$I(x_{t};{\hat{x}}_{t+1})$
 will increase to be closed to 
$I(x_{t};x_{t+1})$
.

Third, according to Theorem 2, 
$I\left(y_{t};y\left(t+1\right)\right)=I\left(x_{t},{\hat{x}}_{t+1}\right)$
 will also be increased such that it can be closed to 
$I(x_{t};x_{t+1})$
.

Because the macro-dynamics is the information bottleneck of the entire channel, therefore its information must be increased as training. At the same time, the determinant of the Jacobian of 
$\psi_{\alpha }$
 and the entropy of 
$y_{t}$
 will also be increased in a general case. This conclusion is implied in Theorem 4.

**Theorem** **4.***(Information on bottleneck is the lower bound of the encoder): For the squeezed information channel shown in Figure 3, the determinant of the Jacobian matrix of 
$\psi_{\alpha }$
 and the Shannon entropy of 
$y_{t}$
 are lower bounded by the information of the entire channel:*

(23)
$$H\left(x_{t}\right)+E(\ln|\det\left(J_{\psi_{\alpha }}\left(x_{t}\right)\right)\left|\right)\geq H\left(y_{t}\right)+E(\ln|\det\left(J_{\psi_{\alpha },y_{t}}\left(x_{t}\right)\right)\left|\right)\geq I\left(x_{t};{\hat{x}}_{t+1}\right),$$

*where H is the Shannon entropy measure, 
$J_{\psi_{\alpha }}\left(x_{t}\right)$
 is the Jacobian matrix of the bijector 
$\psi_{\alpha }$
 at the input 
$x_{t}$
, and 
$J_{\psi_{\alpha },y_{t}}\left(x_{t}\right)$
 is the sub-matrix of 
$J_{\psi_{\alpha }}\left(x_{t}\right)$
 on the projection 
$y_{t}$
 of 
$x_{t}^{\prime }$
.*

The proof is also given in Appendix D.

Because the distribution of 
$x_{t}$
 and its Shannon entropy are given, thus, Theorem 4 states that the expectation of the logrithim of 
$\left|\det(J_{\psi_{\alpha }}\left(x_{t}\right))\right|$
 and the entropy of 
$y_{t}$
 must be larger than the information of the entire information channel.

Therefore, once the initial values of 
$E|\det(J_{\psi_{\alpha }}\left(x_{t}\right))|$
 and 
$y_{t}$
 are small, as the model is trained, the mutual information of the entire channel increases, the determinant of the Jacobian must also be increased, and the distribution of the macro-state 
$y_{t}$
 must be more disperse. However, these may not happen if the information 
$I(x_{t};{\hat{x}}_{t+1})$
 has been closed to 
$I(x_{t};x_{t+1})$
 or 
$E|\det(J_{\psi_{\alpha }}\left(x_{t}\right))|$
 and 
$H(y_{t})$
 have been already large enough.

#### 4.1.3. The Effective Information Is Mainly Determined by the Bijector

The previous analysis is about the mutual information but not the effective information of the macro-dynamic which is the key ingredients about causal emergence. Actually, with the good properties of the squeezed information channel, we can write down an expression of the 
EI
 for the macro-dynamic but without the explicit form of it. Accordingly, we find the major ingredient to determine causal emergence is the bijector 
$\psi_{\alpha }$
.

The proof is detailed in Appendix D.

**Theorem** **5.***(The mathematical expression for effective information of the macro-dynamics): Suppose the probability density of 
$x_{t+1}$
 under given 
$x_{t}$
 can be described by a function 
$Pr\left(x_{t+1}|x_{t}\right)\equiv G\left(x_{t+1},x_{t}\right)$
, and the Neural Information Squeezer framework is well trained, then the effective information of the macro-dynamics of 
$f_{\beta }$
 can be calculated by:*

(24)
$$EI_{L}\left(f_{\beta }\right)=\frac{1}{{\left(2L\right)}^{p}}\cdot \int_{\sigma }\int_{R^{p}}G\left(y,\psi_{\alpha }^{-1}\left(x\right)\right)\ln\frac{{\left(2L\right)}^{p}G\left(y,\psi_{\alpha }^{-1}\left(x\right)\right)}{\int_{\sigma }G\left(y,\psi_{\alpha }^{-1}\left(x^{\prime }\right)\right)dx^{\prime }}dydx,$$

*where 
$\sigma \equiv {\left[-L,L\right]}^{p}$
 is the integration region for 
x
 and 
$x^{\prime }$
.*

#### 4.1.4. Change with the Scale (*Q*)

According to Theorems 2 and 3, we have the following Corollary 1:

**Corollary** **1.**
*(The mutual information of macro-dynamics will not change if the model is well trained): For the well trained NIS model, the Mutual Information of the macro-dynamics 
$f_{\beta }$
 will be irrelevant of all the parameters, including the scale q.*


If the neural networks are well-trained, the mutual information on the macro-dynamics will approach to the information in the data 
$\left\{x_{t}\right\}$
. Therefore, no matter how small *q* is (or how large is the scale), the mutual information of the macro-dynamics 
$f_{\beta }$
 will keep constant.

It seems that the scale *q* is an irrelevant parameter on causal emergence. However, according to Theorem 6, smaller *q* will lead to the encoder carrying more effective information.

**Theorem** **6.***(Narrower is Harder): If the dimension of 
$x_{t}$
 is p, then for 
$0<q_{1}<q_{2}<p$
:*

(25)
$$I\left(x_{t};{\hat{x}}_{t+1}\right)\leq I\left(x_{t};y_{t}^{q_{1}}\right)\leq I\left(x_{t};y_{t}^{q_{2}}\right),$$

*where 
$y_{t}^{q}$
 denotes the q-dimensional vector 
$y_{t}$
.*

The mutual information in Theorem 6 is about the encoder, i.e., the micro-state 
$x_{t}$
 and the macro-state 
$y_{t}$
 in different dimension *q*. The theorem states that as *q* decreases, the mutual information of the encoder part must also decrease and more closed to the information limitation 
$I\left(x_{t};{\hat{x}}_{t+1}\right)≃I\left(x_{t};x_{t+1}\right)$
. Therefore, the entire information channel becomes narrower, the encoder must carry more useful and effective information to transfer to the macro-dynamics. In addition, the prediction becomes harder.

### 4.2. Empirical Results

We test our model on several data sets. All the data are generated by the simulated dynamical models. In addition, the models include continuous dynamics and discrete Markovian dynamics.

#### 4.2.1. Spring Oscillator with Measurement Noise

The first experiment to test our model is a simple spring oscillator following the dynamical equations:
(26)
$$\left\{\begin{array}{c} dz/dt=v\\ dv/dt=-z \end{array}\right.$$

where *z* and *v* are position and velocity of the oscillator in one dimension, respectively. The states of the system can be represented as 
x=(z,v)
.

However, we can only observe the state from two sensors with measurement errors. Suppose the observational model is

(27)
$$\left\{\begin{array}{c} {\tilde{x}}_{1}=x+\zeta \\ {\tilde{x}}_{2}=x-\zeta \end{array}\right.$$

where 
ζ∼N(0,σ)
 is a random number following two dimensional Gaussian distribution, and 
σ
 is the vector of the standard deviations for position and velocity. In this example, we can understand the states 
x
 as latent macro-states and the measurements 
${\tilde{x}}_{1},{\tilde{x}}_{2}$
 are micro-states. What the NIS will do is recover the latent macro-state 
x
 from the measurements.

According to Equation (Equation 27), although there is noise to disturb the measurement of the state, it can be easily eliminated by adding the measurements on the two channels together. Therefore, if NIS can discover a macro-state which is the addition of the two measurements, then it can easily obtain the correct dynamics. We sample the data for 10,000 batches (with Euler method and 
dt=1
), and in each batch, we randomly generate 100 random initial states and perform one step dynamic to obtain the state at the next time step. We use these data to train the neural network. To compare, we also use the same data set to train an ordinary feed-forward neural network with the same number of parameters.

The results are shown in Figure 4. To test if NIS can learn the real latent macro state, we directly plot the predicted and the real latent states. As shown in Figure 4a, the predicted and the real curves collapse together which means NIS can recover the macro state in the data although it is unknown. As a comparison, the feed-forward neural network cannot recover the macro state. We can also check if the NIS can learn the dynamic of the macro states by plotting the derivatives of the states (
dz/dt,dv/dt
) against the macro state variables (
v,z
). If the learned dynamics follows Equation (Equation 26), then two cross-over lines for 
dz/dt=v
 and 
dv/dt=−z
 can be observed as shown in Figure 4c. However, the same pattern cannot be reproduced on the common feed-forward network as shown in Figure 4d. We also test the well-trained NIS by multiple-step prediction as shown in Figure 4e. Although there are larger and larger deviation from the prediction and the real data, the general trends can be captured by NIS model. We further study how the dimension averaged causal emergence 
dCE
 changes with the scale *q* which is measured by the number of effective information channels on the well-trained NIS model as shown in Figure 4f. 
dCE
 peaks at 
q=2
 which is exactly same as in the ground truth.

Further, we use experimental results to verify the theorems mentioned in the previous section and the theory of information bottleneck [45]. First, we show how the mutual information of 
$I(x_{t},x_{t+1})$
, 
$I(y_{t},y\left(t+1\right))$
, and 
$I(x_{t},{\hat{x}}_{t+1})$
 change with time (epoch) when *q* takes different values as shown in Figure 5c,d. The results show that all the mutual information converge as predicted by Theorems 2 and 3. We also plot the mutual information between 
$x_{t}$
 and 
$y_{t}$
 with different *q* to test Theorem 6, and the results show that the mutual information increases when *q* increases as shown in Figure 5a.

According to the information bottleneck theory [45], the mutual information between latent variable and output may increase while the information between input and latent variable should increase in the early stage and then decrease as training process proceed. As shown in Figure 5b, this conclusion is confirmed by the NIS model where the macro-states 
$y_{t}$
 and the prediction 
y(t+1)
 are all latent variables. Although the same conclusion is obtained, the information bottleneck can be reflected by the architecture in NIS model much clearer than the general neural networks because 
$y_{t}$
 and 
y(t+1)
 is the bottleneck and all other irrelevant information is discarded by the variable 
$x_{t}^{\prime \prime }$
 as shown in Figure 3.

#### 4.2.2. Simple Markov Chain

In the second example, we show NIS can work on discrete Markov chain, and the coarse-graining strategy can work on state space. The Markov chain to generate the data is the following probability transition matrix:
(28)
$$\left(\begin{array}{cccccccc} 1/7 & 1/7 & 1/7 & 1/7 & 1/7 & 1/7 & 1/7 & 0\\ 1/7 & 1/7 & 1/7 & 1/7 & 1/7 & 1/7 & 1/7 & 0\\ 1/7 & 1/7 & 1/7 & 1/7 & 1/7 & 1/7 & 1/7 & 0\\ 1/7 & 1/7 & 1/7 & 1/7 & 1/7 & 1/7 & 1/7 & 0\\ 1/7 & 1/7 & 1/7 & 1/7 & 1/7 & 1/7 & 1/7 & 0\\ 1/7 & 1/7 & 1/7 & 1/7 & 1/7 & 1/7 & 1/7 & 0\\ 1/7 & 1/7 & 1/7 & 1/7 & 1/7 & 1/7 & 1/7 & 0\\ 0 & 0 & 0 & 0 & 0 & 0 & 0 & 1 \end{array}\right)$$

The system has 8 states, and seven of them can transfer each other. The last state is standalone. We use a one-hot vector to encode the states. Therefore, for example, state 2 will be represented as 
(0,1,0,0,0,0,0,0)
. We sample the initial state for 50,000 batches to generate data. We then feed these one-hot vectors into the NIS framework, after training for 50,000 epochs, we can obtain an effective model. The results are shown in Figure 6.

By systematically search for different *q*, we found that the dimension averaged causal emergence (dCE) peaks at 
q=1
 as shown in Figure 6a. On the optimal scale, we can visualize the coarse-graining strategy by Figure 6b, in which the x-coordinate is the decimal coding for different states, and the y-coordinate represents the coding for the macro-states. We find that the coarse-graining mapping successfully classifies the first seven states into a one macro-state, and leaves the last state stay alone. This learned coarse-graining strategy is identical as the example shown in [6].

We also visualize the learned macro-dynamics as shown in Figure 6c. This is a linear mapping when 
$y_{t}<0$
 and almost a constant for 
$y_{t}>0$
. Therefore, the dynamics can guarantee that all the first seven micro-states can be separated with the last state. We also verify Theorem 2 in Figure 6d.

#### 4.2.3. Simple Boolean Network

Our framework can not only work on continuous time series and Markov chain, but also can work on a networked system in which each node follows a discrete micro mechanism.

For example, boolean network is a typical discrete dynamical system in which the node contains two possible states (0 or 1), and the state of each node is affected by the state of the neighbors connected to it. We follow the example in [5]. Figure 7 shows an exampled boolean network with four nodes, and each node follows the same micro mechanism as shown in the table of Figure 7. In the table, each entry is the probability of each node’s state conditions on the state combination of its neighbors. For example, if the current node is A, then the first entry is 
$Pr(x_{A}^{t+1}=0|x_{C}^{t}=0,x_{D}^{t}=0)=0.7$
, which means that A will take value 0 with probability 0.7 when the state combination of C and D is 00. By taking all the single node mechanisms together, we can obtain a large Markovian transition matrix with 
$2^{4}=16$
 states which is the complete micro mechanism of the whole network.

We sample the one step state transition of the entire network for 50,000 batches and each batch contains 100 different initial conditions which are randomly sampled from the possible state space evenly, and we then feed these data to the NIS model. By systematically search for different *q*, we found that the dimension averaged causal emergence peaks at 
q=1
 as shown in Figure 8a. Under this condition, we can visualize the coarse-graining strategy by Figure 8b, in which the x-coordinate is the decimal coding for the binary micro-states (e.g., 5 denotes for the state 0101), and the y-coordinate represents the codes for macro-states. The data points can be clearly classified into four clusters according to their y-coordinate. This means the NIS network found four discrete macro-states although the states are continuous real numbers. Interestingly, we found that the mapping between the 16 micro states and four macro states are identical as the coarse-graining strategy shown in the example in Ref. [5]. However, any prior information neither the method on how to group the nodes nor the coarse graining strategy, nor the dynamics are known by our algorithm. Finally, Theorems 2 and 6 are verified in this example as shown in Figure 8c,d.

## 5. Concluding Remarks

In this paper, we propose a novel neural network framework, Neural Information Squeezer, for discovering coarse-graining strategy, macro-dynamic and emergent causality in time series data. We first define effective coarse-graining strategy and macro-dynamic by constraining the coarse-graining strategies to predict the future micro-state with a precision threshold. Then, the causal emergence identification problem can be understood as a maximization problem for effective information under the constraint.

We then use an invertible neural network incorporating with the projection operation to realize the coarse-graining strategy. The usage of invertible neural network can not only allow us to reduce the number of parameters by sharing them between the encoder and the decoder but also can facilitate us to analyze the mathematical properties of the whole NIS architecture.

By treating the framework as a squeezed information channel, we can prove four important theorems. The results show that if the causal connection in the data is strong, then as we train the neural networks, the macro-dynamics will increase its informativeness. In addition, during this process, the determinant of the Jacobian of the bijector will increase at the same time. We also found a mathematical expression for the effective information of the macro-dynamics without the explicit dependence on the macro-dynamics, and it is determined solely by the bijector and the data when the whole framework is well trained. Furthermore, if the framework has been trained in a sufficient time, the mutual information of the macro-dynamics will keep a constant no matter the scale *q* is. However, as *q* decreases, the mutual information or the bandwidth on the encoder part also decreases and closed to the information limitation on the entire channel such that it can make correct prediction for the future micro-states. Thus, the task becomes harder for the encoder because more effective information must be encoded and pass to the dynamics learner such that it can make correct prediction with less information. Numerical experiments show that our framework can reconstruct the dynamics in different scales and also can discover emergent causality in data on several classic causal emergence examples.

There are several weak points in our framework. First, it can only work on small data set. The major reason is the invertible neural network is very difficult to train on large data set. Therefore, we will use some special techniques to optimize the architecture in future. Second, the framework is still lack of explainability, the grouping method for variables is implicitly encoded in the invertible neural network although we can illustrate what the coarse-graining mapping is, and decompose it into information conversion and information discarding parts clearly. A more transparent neural network framework with more explanatory power is deserved for future studies. Third, the conditional distribution that the model can predict actually is limited as Gaussian or Laplacian, and it should be extended to more general distributional forms in future studies.

There are several theoretical problems left for future studies. For example, we conjecture that all coarse-graining strategies can be decompose into a bijection and a projection, but this needs strict mathematical proof. Second, although an explicit expression for EI on macro-dynamics has been derived under NIS, we still cannot directly predict the causal emergence in the data. We believe that a more concise analytic results on the EI should be derived by setting some constraints on the data. Furthermore, we think the meaning and the usage of the discarding variable 
$x_{t}^{\prime \prime }$
 should be further explored since that it may relate with the redundant information of a pair of variables toward a target [46]. Therefore, we guess more deep connections between the framework of NIS and the mutual information decomposition may exist and NIS may work as a numeric tool to decompose the mutual information.

## Figures and Tables

**Figure 1 entropy-25-00026-f001:**
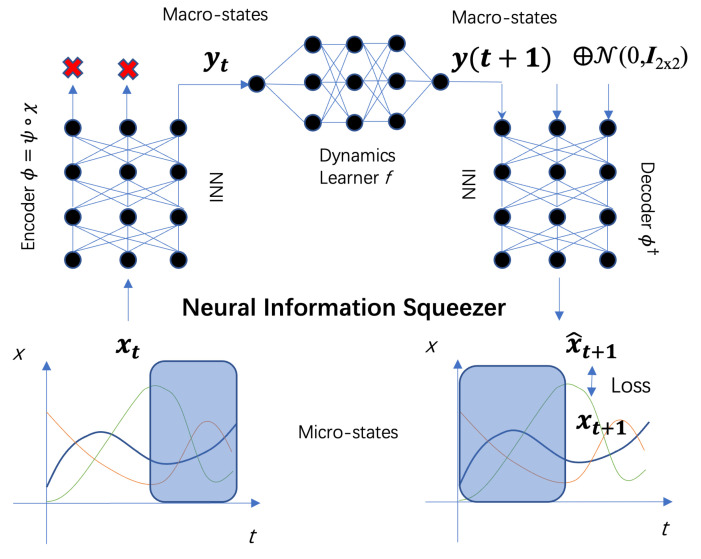
The workflow and the framework of the neural information squeezer. 
$x_{t}$
 is the data at time *t*. Encoder 
$\psi_{\alpha }$
 is an invertible neural network (INN), from which the coarse-grained data 
$y_{t}$
 can be generated. The dynamics learner 
$f_{\beta }$
 is a common feed-forward neural network with parameters 
β
. Through it the evolution from 
$y_{t}$
 to 
y(t+1)
 can be conducted. The decoder converts the predicted macro-state of the next time step 
y(t+1)
 into the prediction of the micro-state at the next time step 
${\hat{x}}_{t+1}$
.

**Figure 2 entropy-25-00026-f002:**
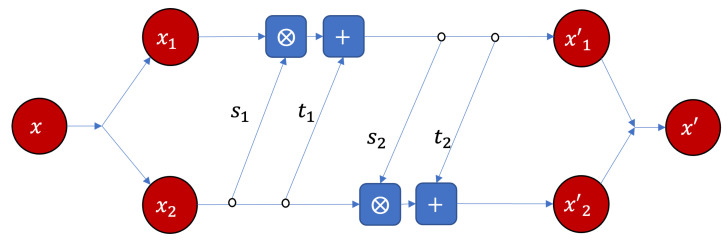
The RealNVP neural network implementation of the basic module of the bijector 
ψ
, where 
$s_{1},s_{2}$
 and 
$t_{1},t_{2}$
 are all feed-forward neural networks with three layers, 64 hidden neurons, and ReLU active function. 
$s_{i}$
s and 
$t_{i}$
s share parameters, respectively. ⨂ and + represent element-wised product and addition, respectively. 
$x=x_{1}⨁x_{2}$
 and 
$x^{\prime }=x_{1}^{\prime }⨁x_{2}^{\prime }$
.

**Figure 3 entropy-25-00026-f003:**
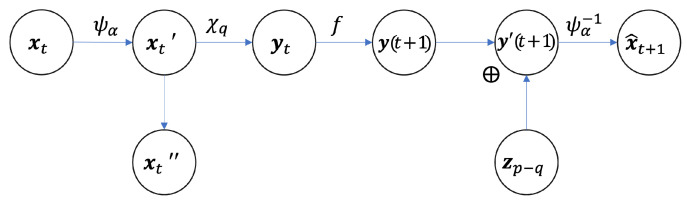
The graphic model of the neural information squeezer as a squeezed information channel.

**Figure 4 entropy-25-00026-f004:**
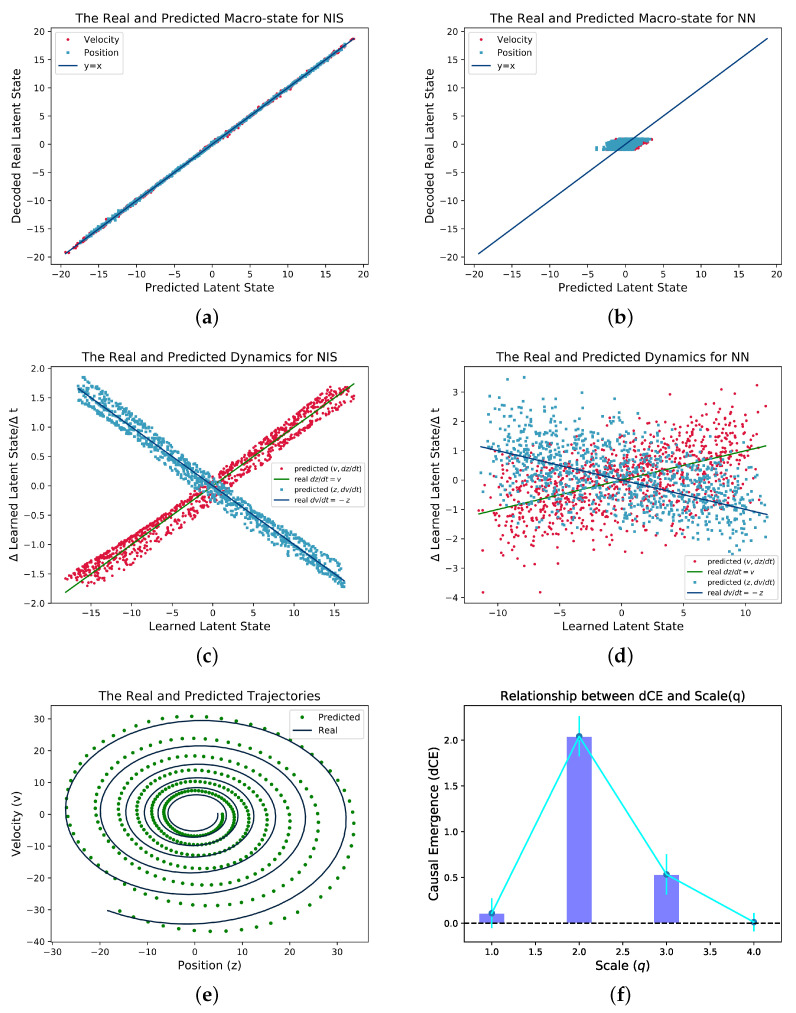
Experimental Results for the Simple Spring Oscillator with Measurement Noise. We sample data from Equations (Equation 26) and (Equation 27), and we use Euler method to simulate by taking 
dt=0.1
. (**a**,**b**) show the real macro-state versus the predicted ones both for NIS and the ordinary feed-forward neural network, respectively; (**c**,**d**) show the real and predicted dynamics, i.e., the dependence between 
dz/dt
 and *v*, and 
dv/dt
 with *z* for both neural networks for comparison; (**e**) shows the real and predicted trajectories with 400 time steps starting from the same latent state; (**f**) shows the dependence of the dimension averaged of 
CausalEmergence
 (dCE) on *q* (the number of effective channels).

**Figure 5 entropy-25-00026-f005:**
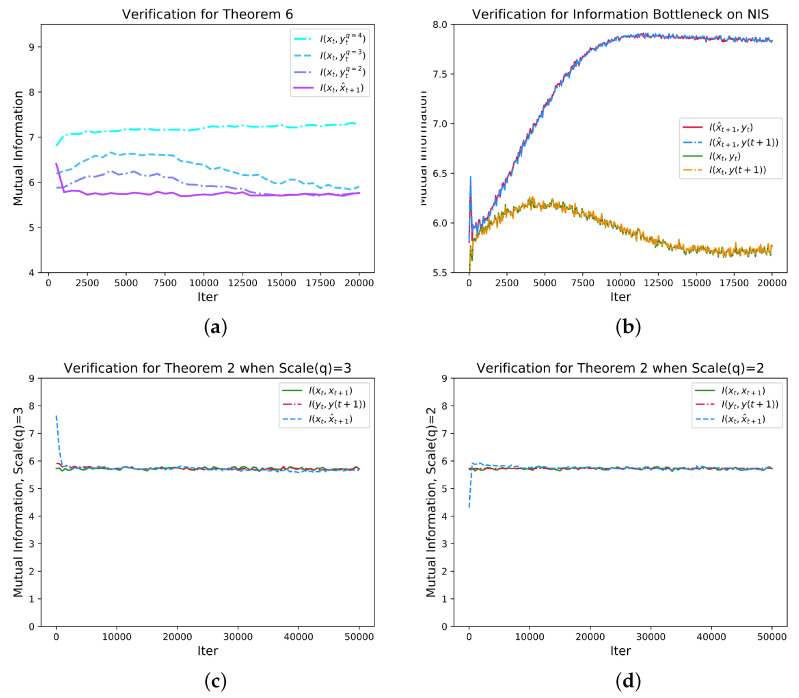
Various mutual information between variables change with training iterations. (**a**) shows the change of mutual information 
$I(x_{t},y_{t}^{q=4})$
, 
I(xt,ytq=3)
, 
$I(x_{t},y_{t}^{q=2})$
 and 
$I(x_{t},{\hat{x}}_{t})$
 with the increase of the number of iterations. From the figure, we can see that within the specified number of iterations, 
$I\left(x_{t},{\hat{x}}_{t}\right)\leq I\left(x_{t},y_{t}^{q=2}\right)\leq I\left(x_{t},y_{t}^{q=3}\right)\leq I\left(x_{t},y_{t}^{q=4}\right)$
. Among them, *q* is the dimension of the coarse-grained system. (**b**) verifies the theory of information bottleneck on NIS when Scale (q) = 2. (**c**,**d**) show the change of mutual information 
$I(x_{t},x_{t+1})$
, 
I(yt,yt+1)
 and 
$I(x_{t},{\hat{x}}_{t})$
 with the increase of the number of iterations. It can be seen that under different scales, the three mutual information values are close to each other. The standard deviations of the three variables gradually decrease with iteration. Therefore, 
$I\left(x_{t},x_{t+1}\right)\approx I\left(y_{t},y_{t+1}\right)=I\left(x_{t},{\hat{x}}_{t}\right)$
 is reflected.

**Figure 6 entropy-25-00026-f006:**
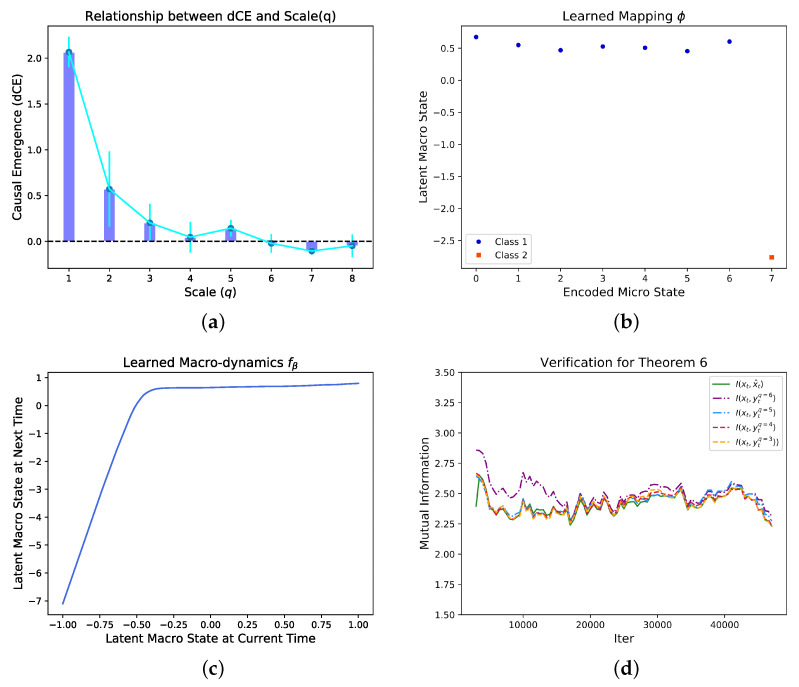
The dependence of the dimension averaged 
CausalEmergence
 (dCE) on different scales (*q*) of the Markov dynamics (**a**), the learned mapping between micro states and macro states on the optimal scale (*q*) (**b**), and the learned macro-dynamics the mapping from 
$y_{t}$
 to 
y(t+1)
 (**c**). There are two clear separated clusters on the *y*-axis in (**b**) which means the macro states are discrete. We found that the two discrete macro states and the mapping between micro and discrete macro states are identical as the example in Ref. [6] which means the correct coarse-graining strategy can be discovered by our algorithm automatically under the condition without any prior information. In (**d**), 
$I\left(x_{t},{\hat{x}}_{t}\right)\leq I\left(x_{t},y_{t}^{q=3}\right)\leq I\left(x_{t},y_{t}^{q=4}\right)\leq I\left(x_{t},y_{t}^{q=5}\right)\leq I\left(x_{t},y_{t}^{q=6}\right)$
 is reflected. In order to make the data clearer, we have taken a moving average for each group of data. This result can be regarded as the verification of Theorem 6.

**Figure 7 entropy-25-00026-f007:**
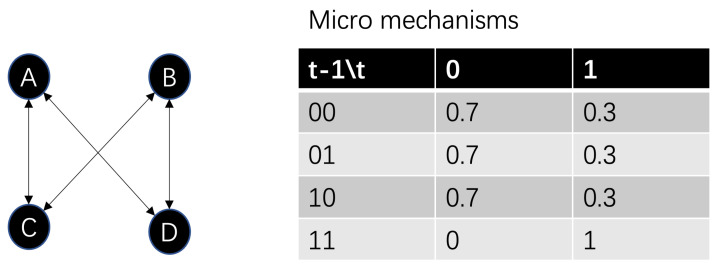
An exampled Boolean network (**left**) and its micro mechanisms on nodes (**right**). Each node’s state on the next time step is affected by its neighboring nodes’ state combination randomly. The transition probabilities (micro mechanisms) on each case are shown in the table.

**Figure 8 entropy-25-00026-f008:**
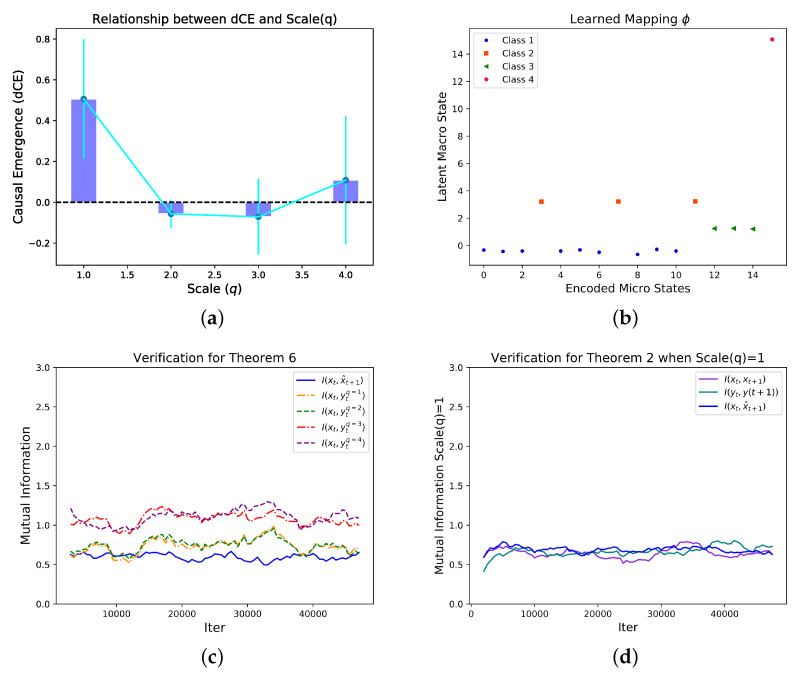
Experimentsl Results for the Boolean Network. The dependence of the dimension averaged 
CausalEmergence
 (dCE) on different scales (*q*) (**a**) and the learned mapping between micro states and macro states on the optimal scale (*q*) (**b**). There are four clear separated clusters on the *y*-axis in (**b**) which means the macro states are discrete. We found that the four discrete macro states and the mapping between micro and discrete macro states are identical as the example in Ref. [5] which means the correct coarse-graining strategy can be discovered by our algorithm automatically under the condition without any prior information. (**c**) shows the change of mutual information 
$I(x_{t},y_{t}^{q=4})$
, 
I(xt,ytq=3)
, 
$I(x_{t},y_{t}^{q=2})$
,
$I(x_{t},y_{t}^{q=1})$
 and 
$I(x_{t},{\hat{x}}_{t})$
 with the increase of the number of iterations. From the figure, we can see approximately that within the specified number of iterations, 
$I\left(x_{t},{\hat{x}}_{t}\right)\leq I\left(x_{t},y_{t}^{q=1}\right)\leq I\left(x_{t},y_{t}^{q=2}\right)\leq I\left(x_{t},y_{t}^{q=3}\right)\leq I\left(x_{t},y_{t}^{q=4}\right)$
. (**d**) shows the change of mutual information 
$I(x_{t},x_{t+1})$
, 
I(yt,yt+1)
 and 
$I(x_{t},{\hat{x}}_{t})$
 with the increase of the number of iterations. In order to easily observe the trend of data changes, we added a moving average curve for each group of data. It can be seen that under different scales, the three mutual information values are close to each other. 
$I\left(x_{t},x_{t+1}\right)\approx I\left(y_{t},y_{t+1}\right)=I\left(x_{t},{\hat{x}}_{t}\right)$
 is reflected. Considering the experimental error, the overall trend of the data still conforms to the theorem.

## Data Availability

All the codes and data are available at: https://github.com/jakezj/NIS_for_Causal_Emergence, accessed on 24 October 2022.

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
