# Peer review of "Neural Information Squeezer for Causal Emergence"

_entropy, 2022, doi:10.3390/e25010026_

Round 1

Reviewer 1 Report

This is quite an interesting paper and development in the space of automating the detection of informative / causal emergence in real datasets. I think the structure, results, and overall approach here is quite strong.

I think there’s still some additional motivation that could help bolster the rationale for why 1) this technique is useful and different from other approaches to the same problem and 2) zooming out—why causal emergence in general is meaningful to search for in data (e.g. see potentially useful citations of work described in [1,2,3,4,5] below). If possible, I’d love to see the authors spend a little extra time in the introduction motivating the utility of this framework in general.

The caption of Figure 1 should be expanded upon, fully detailing what each panel describes.

Also, one thing that I think would help broaden the audience for this work is to extend e.g. Figure 3 to include a step-by-step example with not just variable names but also actual values. This would then lead smoothly into the examples later in Section 3.

Note: I played around a bit with the code, and it’s quite a neat implementation! Congrats.

Last small point — I believe the most appropriate citation for the algorithms used in citation #11 in the manuscript (Klein & Hoel, 2020) is [6] below.

[1] Swain, A., Williams, S. D., Di Felice, L. J., & Hobson, E. A. (2022). Interactions and information: exploring task allocation in ant colonies using network analysis. Animal Behaviour, 189, 69-81. doi: 10.1016/j.anbehav.2022.04.015.

[2] Varley, T. F. (2022). Flickering emergences: The question of locality in information-theoretic approaches to emergence. arXiv preprint arXiv:2208.14502.

[3] Klein, B., Hoel, E., Swain, A., Griebenow, R., & Levin, M. (2021). Evolution and emergence: higher order information structure in protein interactomes across the tree of life. Integrative Biology, 13(12), pp. 283–294. doi: 10.1093/intbio/zyab020.

[4] Varley, T.F. & Hoel, E. (2022). Emergence as the conversion of information: a unifying theory. Phil. Trans. R. Soc. A. 38020210150 doi: 10.1098/rsta.2021.0150.

[5] Ravi, D., Hamilton, J. L., Winfield, E. C., Lalta, N., Chen, R. H., & Cole, M. W. Causal emergence of task information from dynamic network interactions in the human brain. Rev. Neurosci, 31, 25-46.

[6] Klein, B., Swain, A., Byrum, T., Scarpino, S. V. & Fagan, W. F. (2022). Exploring noise, degeneracy and determinism in biological networks with the einet package. Methods in Ecology and Evolution, 13, 799– 804. doi: 10.1111/2041-210X.13805.

Author Response

Dear referee,

Thank you very much for your review of our paper and recognition of its practical significance. At the same time, we are grateful for your criticism of some problems in our paper description.

Our responses and modifications to the problems are as follows:

Problem 1:

I think there’s still some additional motivation that could help bolster the rationale for why 1) this technique is useful and different from other approaches to the same problem and 2) zooming out—why causal emergence in general is meaningful to search for in data (e.g. see potentially useful citations of work described in [1,2,3,4,5] below). If possible, I’d love to see the authors spend a little extra time in the introduction motivating the utility of this framework in general.

Response:

We have read and quoted several papers provided by you in the introduction. This can make our discussion more relevant to existing research and tells readers more application directions. Thank you very much for your ideas and relevant literature.

In paragraph 1 page 1, we added that “Previous works have shown that causal emergence can be applied in wide areas including studies on ant colony[12], protein interactomes [13], brain[14], and biological networks[15].”

Paragraph 3 is our new content: “The problem is difficult because the search space is all possible mapping functions between micro and macro, which is huge. To solve this problem, Klein et al. focuses on the complex systems with network structures[ 15, 16 ], and converted the problem of coarse-graining into node clustering. That is, to find a way to group nodes into clusters such that the connections on the cluster level has larger EI than the original network. This method has been widely applied in various areas[ 12 –14 ], nevertheless, it assumes that the underlying node dynamic is diffusion(random walks). While, real complex systems have much richer node dynamics. For a general dynamic, even if the node grouping is given the coarse-grained strategy still needs to consider how to map the micro-states of all nodes in a cluster to the macro-state of the cluster[5]. The tedious searching on a huge functional space of coarse-graining strategies is also needed.”

The article corresponding to the serial number is in the coverletter.

12. Swain, A.; Williams, S.D.; Di Felice, L.J.; Hobson, E.A. Interactions and information: exploring task allocation in ant colonies using network analysis. Animal Behaviour 2022, 189, 69–81.
13. Klein, B.; Hoel, E.; Swain, A.; Griebenow, R.; Levin, M. Evolution and emergence: higher order information structure in protein interactomes across the tree of life. Integrative Biology 2021, 13, 283–294

14. Ravi, D.; Hamilton, J.L.; Winfield, E.C.; Lalta, N.; Chen, R.H.; Cole, M.W. Causal emergence of task information from dynamic network interactions in the human brain. Rev. Neurosci 2022, 31, 25–46.
15. Klein, B.; Swain, A.; Byrum, T.; Scarpino, S.V.; Fagan, W.F. Exploring noise, degeneracy and determinism in biological networks with the einet package. Methods in Ecology and Evolution 2022, 13, 799–804.

Problem 2:

The caption of Figure 1 should be expanded upon, fully detailing what each panel describes.

Response:

We have added a more complete description to Figure 1, so that we can more clearly recognize the specific role of each panel.

Problem3:

Also, one thing that I think would help broaden the audience for this work is to extend e.g. Figure 3 to include a step-by-step example with not just variable names but also actual values. This would then lead smoothly into the examples later in Section 3.

Response:

Figure 3 is not an illustration of the algorithm steps, but an abstract representation of the probabilistic graph model in Figure 1, which is used to prove a series of propositions such as theorem 2. Therefore, it is not changed.

Problem4:

Last small point — I believe the most appropriate citation for the algorithms used in citation #11 in the manuscript (Klein & Hoel, 2020) is [6] below.

Response:

We added the most appropriate citation for the algorithms. For more details, please refer to the revised manuscript.

Your suggestions are of great help to the improvement of our paper. Apart from your suggestion, we proofread some other content. I hope you can review our content again. If there are other problems, we are willing to accept criticism and correction again.

Sincerely,

Jiang Zhang & Kaiwei Liu

Reviewer 2 Report

this is an interesting submission that tackles and important problem, that is how to extract macro variables (and their causal structure) from micro-variables observations.

In general I think the paper is well written and sound, but I did not checked the proof in details.

I have just some problem with some of the definitions and results, in particular their formulations:

- Definition 2, there are no assumption whatsoever on the function\phi_q ?

- Definition 3, "is closed to the macro-states as possible as we can" is not appropriate for a definition. I suggest to write down the definition of macro-state dynamics as the results of an optimization problem for the any-vector norm. Another problem here, it seems then that the macro-state dynamics depends on the chosen norm. Also, given a norm, the dynamic is unique? Also since ||y_t - y(t)||  is a function of t we need to chose how to compare it, uniformly in time ( ||y_t - y(t)|| < \eps ) or on average \sum_t ||y_t - y(t)|| ? ...

- Definition 4, "for a given small real number" does not mean anything ("small" is not defined), I suggest something like defining the \epsilon-effective coarse-graining strategy, also this definition depend on the chosen norm and this should be observed. Additionally, "and the derived macro dynamic is also effective"  what does it mean ?  it is not defined what it means for a macro-dynamic to be effective.

- In Theorem 3 what it means to be "well trained" ?

-

Author Response

Dear referee,

Thank you very much for your review of our paper and recognition of its practical significance. At the same time, we are grateful for your criticism of some problems in our paper description.

Our responses and modifications to the problems are as follows:

Problem 1:

- Definition 2, there are no assumption whatsoever on the function\phi_q ?

Response:

For the assumption on the function , we added some details that , a q dimensional coarse-graining strategy, is a continuous and differential function to map the micro-state  to a macro-state .

Problem 2:

- Definition 3, "is closed to the macro-states as possible as we can" is not appropriate for a definition. I suggest to write down the definition of macro-state dynamics as the results of an optimization problem for the any-vector norm. Another problem here, it seems then that the macro-state dynamics depends on the chosen norm. Also, given a norm, the dynamic is unique? Also since ||y_t - y(t)||  is a function of t we need to chose how to compare it, uniformly in time ( ||y_t - y(t)|| < \eps ) or on average \sum_t ||y_t - y(t)|| ? ...

Response:

We modified Definition 3, the definition of macro-state dynamics. This definition explains in more detail the principle and test method of dynamic systems in the macro state on the basis of time series of macro-states .

Problem 3:

- Definition 4, "for a given small real number" does not mean anything ("small" is not defined), I suggest something like defining the \epsilon-effective coarse-graining strategy, also this definition depend on the chosen norm and this should be observed. Additionally, "and the derived macro dynamic is also effective"  what does it mean ?  it is not defined what it means for a macro-dynamic to be effective.

Response:

In order to better understand the validity of the model, we added a new concept of -effective q coarse-graining strategy and macro-dynamcis in Definition 4. This can define what it means for a macro-dynamic to be effective.

Problem 4:

- In Theorem 3 what it means to be "well trained" ?Response:

Response:

For the means of "well-trained", we explained that mutual information of the model will be closed to the data for a well-trained frame-work in Theorem 3. "well-trained" means for any , the KL-divergence between  and  approaches 0 when the training epoch .

We can see more details about the revision of relevant theorems and definitions in the revised article. Your suggestions are of great help to the improvement of our paper. Apart from your suggestion, we proofread some other content. I hope you can review our content again. If there are other problems, we are willing to accept criticism and correction again.

Sincerely,

Jiang Zhang & Kaiwei Liu
